# Peripheral bone structure, geometry, and strength and muscle density as derived from peripheral quantitative computed tomography and mortality among rural south Indian older adults

**Guru Rajesh Jammy**[1,2], **Robert M. Boudreau**[1], **Iva Miljkovic**[1], **Pawan Kumar Sharma**[2], **Sudhakar Pesara Reddy**[1,2], **Susan L. Greenspan**[3], **Anne B. Newman**[1], **Jane A. Cauley**[1]*

**1** Department of Epidemiology, Graduate School of Public Health, University of Pittsburgh, Pittsburgh, Pennsylvania, United States of America, **2** SHARE INDIA–Mediciti Institute of Medical Sciences, Medchal District, Telangana, India, **3** Department of Medicine, University of Pittsburgh, Pittsburgh, Pennsylvania, United States of America

* jcauley@pitt.edu

## Abstract

Multiple studies have observed a relationship of bone mineral density (BMD) measured by Dual energy X-ray absorptiometry (DXA) and mortality. However, areal BMD (aBMD) measured by DXA is an integrated measure of trabecular and cortical bone and does not measure the geometry of bone. Peripheral Quantitative Computed Tomography (pQCT) provides greater insights on bone structure, geometry and strength. To examine whether higher bone phenotypes and muscle density as measured by pQCT are associated with a lower all-cause mortality, we studied 245 men and 254 women (all age >60) recruited in the Mobility and Independent Living among Elders Study in rural south India. Cox proportional hazards models estimated hazard ratios (HR [95% Confidence Intervals]). After an average follow-up of 5.3 years, 73 men and 50 women died. Among men, trabecular volumetric bone mineral density (vBMD) of radius (HR per SD increase in parameter = 0.59 [0.43, 0.81]) and tibia (0.60[0.45, 0.81]), cortical vBMD of radius (0.61, [0.47, 0.79]) and tibia (0.62, [0.49, 0.79]), cortical thickness of radius (0.55, [0.42, 0.7]) and tibia (0.60, [0.47, 0.77]), polar strength strain index (SSIp) of tibia (0.73 [0.54, 0.98]), endosteal circumference of radius (1.63, [1.25, 2.12]) and tibia (1.54, [1.19, 1.98]) were associated with all-cause mortality. Muscle density (0.67, [0.51, 0.87]) was associated with lower mortality in men. Among women cortical vBMD of radius (0.64, [0.47, 0.87]) and tibia (0.60 [0.45, 0.79]), cortical thickness of radius (0.54, [0.37, 0.79]) and tibia (0.43, [0.30, 0.61]), SSIp of radius (0.70 [0.48,1.01]) and tibia (0.58 [0.37, 0.90]) and endosteal circumference of radius (1.33 [0.97, 1.82]) and tibia (1.83, [1.37, 2.45]) were associated with all-cause mortality. Among men, gait speed mediated the association of muscle density and mortality but there was no mediation for any bone parameters. Conclusion: pQCT bone measures and muscle density were independently associated with mortality among rural south Indian elders.

**Data Availability Statement:** All data are in the manuscript and/or supporting information files.

**Funding:** (PSR) The Mobility and Independent Living in Elders Study (MILES) was carried out with institutional support from the University of Pittsburgh Graduate School of Public Health, SHARE INDIA and Fogarty International Center (FIC) D43 TW009078. (JAC) The Osteoporotic Fractures in Men (MrOS) Study is supported by the National Institutes of Health funding. The following institutes provide support: the National Institute on Aging (NIA), the National Institute of Arthritis and Musculoskeletal and Skin Diseases (NIAMS), the National Center for Advancing Translational Sciences (NCATS), and NIH Roadmap for Medical Research under the following grant numbers: U01 AG027810, U01 AG042124, U01 AG042139, U01 AG042140, U01 AG042143, U01 AG042145, U01 AG042168, U01 AR066160, and UL1 TR000128.

**Competing interests:** The authors have declared that no competing interest exist.

## Introduction

Multiple studies have observed a relationship of bone mineral density (BMD) measured by Dual energy X-ray absorptiometry (DXA) and mortality [1–6]. A meta-analysis of 10 prospective studies with 46182 participants from 5 countries (US, Netherlands, Sweden, Australia, and Brazil) with median follow up of 7 years, showed an increased all-cause mortality of 1.2 fold (Hazard Ratio (HR) 1.20; 95%CI 1.09–1.30) per one standard deviation (SD) decrease in BMD [7]. However, areal BMD (aBMD) measured by DXA is an integrated measure of trabecular and cortical bone and does not measure the geometry of bone. Peripheral Quantitative Computed Tomography (pQCT) provides greater insights on bone structure, geometry and strength.

Among older adults, muscle mass alone cannot fully explain the loss of physical function and muscle strength with age suggesting that there are other aspects of muscle quality which may contribute. Myosteatosis, an excess deposit of fat in the skeletal muscles both at intramuscular and intermuscular levels [8, 9], has been linked with decreased muscular function and physical performance [10–12], aging [13], reduced muscle strength [14], increased hip fractures / fragility fractures [15, 16] and increased mortality [17].

Muscle strength measured using grip strength has been extensively studied in relationship to mortality. A meta-analysis of 42 studies with 3,002,203 participants, including one study from India, showed mortality risk of 1.16- fold (95% CI 1.12–1.20) for every 5 kg decrease in grip strength [18]. Gait speed a measure of lower limb strength and function has also been linked with mortality: a pooled analysis of 34,485 community dwelling older adults followed for 6 to 21 years reported that for every 0.1 m/s increase of gait speed survival was improved by 12% (95% CI 10–13%) [19].

Little is known about pQCT derived bone and muscle measures among non-European older adults. The current demographic trends in India predict an increase in the older population, despite a lower life expectancy compared to developed nations and with subsequent higher disability rates. To our knowledge, there have been no studies on the relationship of pQCT bone measures and muscle density with mortality among this high risk population of rural Indian elders.

In this analysis, we tested the hypothesis that higher pQCT bone phenotypes and muscle density are associated with decreased mortality independent of confounding factors and that these associations may differ by sex and be at least partially mediated by grip strength and gait speed.

## Methods

### Study population

The Mobility and Independent Living Among Elders Study (MILES) is a prospective study which enrolled 562 community dwelling men and women aged 60 years and over, from Medchal region of Telangana state of southern India. MILES was designed to determine the prevalence of age-related chronic diseases, disability and to examine the extent of clinical and subclinical disease [20]. The study was approved by institutional review board at the participating institutions and all subjects provided written informed consent.

At baseline, two visits for data collection were conducted. During the first visit (February 2012-November 2012), participants completed questionnaires including information on health status, smoking, alcohol consumption etc., physical performance tests, fasting glucose tests and blood pressure measurements. At the second baseline visit (June 2012-June 2013), pQCT scans were conducted. Of the total 562 participants recruited, 17 died (11 men, 6

women) between the baseline visit 1 and visit 2; 27 participants (15 men and 12 women) were physically not able to come for pQCT measurement; 3 women moved out of the area and 4 participants (3 men and 1 women) refused to continue in the study. The pQCT scans were conducted on 511 participants; of which, 499 had valid scan data (245 men and 254 women).

## pQCT and calibration

pQCT scans on the radius and tibia were performed using the Stratec XCT-2000 (Stratec Medizintechnik, Pforzheim, Germany). Technicians followed a standardized protocol for positioning and scanning of each subject. Scans were taken at 4% and 33% of the length of radius and at 4%, 33% and 66% of the length of tibia. Subject scans were repeated if artefacts due to motion or beam hardening were present. To monitor the stability of the pQCT scanners, a manufacturer supplied cylindrical Quality Assurance (QA) phantom was scanned daily before subject scans were acquired. All pQCT scans were analysed by a single investigator using the manufacturer software package version 6.00 for the XCT scanners. This software provides a suite of segmentation options to quantify total, trabecular and cortical bone properties from each pQCT image. Before each image was analysed, it was checked for artifacts due to motion or beam hardening; scans with artefacts were not analysed. All 4% radius and tibia scans were analysed using the CALCBD option with an automatic gradient search (contour mode 2) applied to segment bone from the soft tissue background and concentric peeling (peelmode 1, 45%) to segment trabecular and cortical bone. Proximal scans acquired at the 33% and 66% limb locations were segments using a fixed threshold of 710 mg/cm3(Cortmode1). Coefficients of variation (CVs) were determined for pQCT scans by replicating measurements on 15 subjects (CV $\leq$ 2.1%).

## pQCT parameters

For this analysis, we focused on the following pQCT parameters: at the 4% site of radius and tibia—trabecular vBMD; at the 33% sites of radius and tibia—cortical vBMD, cortical thickness, endosteal circumference and polar strength strain index (SSIp); at 66% tibia–muscle density. These parameters were chosen because vBMD is an indicator of bone matrix mineralisation and represents the mechanical quality of the solid bone tissue both at the trabecular and cortical sites. Endosteal circumference and cortical thickness represent bone geometry and strength. SSIp predicts the failure load [21, 22] and has been shown to be a good predictor of long bone bending [22]. All these parameters also have age-related changes due to adaption of stress, strain, and load on the bone and fractures [23, 24]. Muscle density serves as a surrogate marker of fat infiltration within the muscle [25] and reflects the compactness of muscle fibers, protein within the muscle and other soft tissues and can be viewed as a proxy measure of muscle quality.

## Mortality assessment

As a part of the MILES cohort data collection, each participant was followed for death through community health volunteers at the village level. Based on the information on death from the community health volunteers, MILES research staff visited the participants' home to ascertain the event of death and administers a verbal autopsy tool.

All deaths until March 31, 2019 were considered in this analysis. There were total of 123 deaths (men 73/245; women 50/254); the maximum time to death was 76.7 months since the baseline pQCT measurement. The average follow-up was 64.2 months (5.3 years).

## Covariates

Information on covariates was collected through interviewer-administered questionnaires. Self-reported health status was categorised as good/excellent and fair/poor/very poor. Smoking status was categorised as current smoker and not a current smoker. Alcohol consumption was categorised as consumes alcohol and does not consume alcohol. Self-report of the history of stroke was recorded. Direct measures of weight using SECCA® scale was recorded. Height was measured using a SECCA® stadiometer. Body Mass Index (BMI) was calculated as body weight in kilograms divided by height in meters squared. Diabetes was categorised as present if glucose levels were ≥126mg/dL (after a minimum of an 8-hour fast), self-report of diabetes or used insulin or hypoglycemic medications. Hypertension was considered present if participant self-reported hypertension, reported use of an anti-hypertensive medication or blood pressure assessment (≥ 140/90 mm of Hg). Activities of daily living (ADL) were assessed using the standard tasks of eating, dressing, bathing, transferring from bed to chair and using the toilet. ADL was categorised as ADL disability if the participant reported difficulty in any one of the tasks. Information on history of fractures (hip, arm, wrist shoulder, spine and any other bones) over the past 5 years was obtained. Serum 25-hydroxy vitamin D levels (ng/ml) were measured using the high-performance liquid chromatography (HPLC) method. Grip strength was measured twice in each hand using a hand-held dynamometer. In this analysis, we used the average of the two readings of the participants' dominant hand. The Short Physical Performance Battery (SPPB) consists of a group of physical measures used to predict disability among older populations [26] and combines gait speed, balance tests and chair stands with a total score ranging from 0 (worst performance) to 12 (best performance). The 4-meter timed walk was performed twice for each participant as a part of the SPPB. Gait speed was calculated as the average in meters per second. All covariate information was collected during the first baseline visit.

## Statistical analysis

All data underlying the findings described in the manuscript are attached as a supplement.

There were significant differences in the pQCT parameters between men and women; hence, sex specific analyses were done. Serum 25-hydroxy vitamin D values were not normally distributed and were log transformed. We described the participant's characteristics at the baseline visit using means ± SD or prevalence. Two-sample t-tests or Wilcoxon rank sum tests (continuous variables) or chi-square tests (categorical variables) were used to compare characteristics between men and women and between survivors and deceased participants. The pQCT parameters were categorized as quartiles and Kaplan-Meier survival curves with log rank tests were conducted. Cox proportional hazards models were used to assess the association between the pQCT measures and all-cause mortality; hazards ratio and 95% confidence intervals were calculated. The covariates of interest were age, height, weight, current smoker, consumes alcohol, self-reported health status, hypertension, diabetes, stroke, 25-hydroxy vitamin D level and ADL disability. The minimally adjusted model (model 1) included age, height and weight and fully adjusted model (model 2), model 1 + smoking, alcohol, health status opinion, hypertension, diabetes, stroke, log 25-hydroxy vitamin D and ADL disability. All models with muscle density were also adjusted for muscle cross sectional area (CSA). The proportional hazards assumption was confirmed by Supremum test.

As grip strength and gait speed are associated with mortality and these are in the causal pathway of bone / muscle and mortality, a causal mediation analysis was conducted. The mediator models were model 3, model 2 + grip strength and, model 4 = model 2 + gait speed. Mediation was considered present if there was attenuation of hazards ratio of more than 10% in the

models with and without the mediator variables (models 3 and 4 compared to model 2). The pQCT parameters in the mediator models which had significant association with all-cause mortality were considered for the causal mediation analysis. The overall proportion of mediation was considered in the causal mediation analysis. Results were considered statistically significant when a p-value was less than 0.05. All statistical analyses were carried out using Stata/IC 13.1 and SAS 9.4 software.

## Results

### Baseline characteristics

The baseline characteristics comparing 245 men and 254 women, and comparing men and women who survived versus deceased are presented in the Table 1. The mean age of the men (68.2 ± 6.62) was similar to women (67.2 ± 6.21). Men compared to women were significantly taller, heavier, had lower BMI, higher waist circumference, lower hip circumference, higher grip strength, faster gait speed, higher SPPB score, higher prevalence of completion of 400-meter walk test, currently smoking and alcohol consumption.

There were 123 deaths (25%) [73 men (30%) and 50 (20%) women among the 499 participants. Men and women who died were significantly older than survivors. Men who died were taller, had lower grip strength, slower gait speed, and lower SPPB scores. Women who died had lower body weight and BMI, higher 25-hydroxy vitamin D levels, slower gait speed, lower grip strength, lower SPPB score and, reported lower health status and a higher prevalence of stroke.

**Table 1. Participants characteristics among men and women of MILES.**

| Baseline characteristics | Total participants | | | Men | | | Women | | |
|---|---|---|---|---|---|---|---|---|---|
| | Men (N = 245) | Women (N = 254) | p value (Men vs Women) | Alive (N = 172) | Deceased (N = 73) | p value (alive vs deceased) | Alive (N = 204) | Deceased (N = 50) | p value (alive vs deceased) |
| Age (years) | 68.2 ± 6.62 | 67.2 ± 6.21 | 0.0749 | 67.3 ± 6.47 | 70.32 ± 6.54 | 0.0003* | 66.59 ± 5.68 | 69.68 ± 7.59 | 0.01* |
| Height (cm) | 160.62 ± 5.6 | 147.01 ± 5.95 | <0.0001* | 160.06 ± 5.3 | 161.92± 6.09 | 0.0209* | 147.37 ± 5.59 | 145.54 ± 7.12 | 0.18 |
| Weight (kg) | 55.87 ± 11.51 | 49.96 ± 12.21 | <0.0001* | 55.98 ± 11.34 | 55.62± 11.97 | 0.682 | 50.81 ± 12.1 | 46.53 ± 12.12 | 0.02* |
| BMI (kg/m$^2$) | 21.58 ± 3.92 | 22.98 ± 4.8 | 0.0045* | 21.79 ± 3.97 | 21.09 ± 3.78 | 0.204 | 23.28 ± 4.85 | 21.76 ± 4.48 | 0.047* |
| Vitamin D (ng/ml) | 30.62± 16.15 | 29.35 ± 17.65 | 0.1752 | 30.73 ± 16.21 | 30.36± 16.11 | 0.69 | 28.58±18.36 | 32.76±13.68 | 0.026* |
| Average Gait speed (m/s) | 0.69 ± 0.18 | 0.58 ± 0.16 | <0.0001* | 0.72 ± 0.18 | 0.62 ± 0.18 | <0.0001* | 0.61 ± 0.15 | 0.46 ± 0.17 | <0.0001* |
| Average Grip strength (kg) | 20 ± 6.04 | 12.45 ± 4.74 | <0.0001* | 20.99 ± 5.91 | 17.68 ± 5.76 | 0.0003* | 12.86 ± 4.52 | 10.76 ± 5.29 | 0.01* |
| SPPB score | 8.78 ± 2.75 | 7.22 ± 2.82 | <0.0001* | 9.39 ± 2.44 | 7.33 ± 2.89 | <0.0001* | 7.74 ± 2.59 | 5.14 ± 2.78 | <0.0001* |
| Follow up time (years) | 5.22 ± 1.66 | 5.47 ± 1.5 | 0.0603 | 6.12 ± 0.33 | 3.12 ± 1.64 | <0.0001* | 6.12 ± 0.32 | 2.84 ± 1.56 | <0.0001* |
| Health status (Good) | 113 (46.1) | 105 (41.3) | 0.2814 | 85 (49.4) | 28 (38.3) | 0.112 | 93 (45.5) | 12 (24) | 0.006* |
| ADL difficulty (at least one activity) | 197 (80.4) | 199 (78.4) | 0.5694 | 135 (78.5) | 62 (85) | 0.245 | 155 (76) | 44 (88) | 0.06 |
| Current smokers | 107 (43.7) | 1 (0.4) | <0.0001* | 70 (41) | 37 (51) | 0.149 | 1 (0.5) | 0 (0) | 0.62 |
| Consumes alcohol | 175 (71.4) | 146 (57.5) | 0.0011v | 123 (71) | 52 (71) | 0.964 | 118 (57.8) | 28 (56) | 0.81 |
| Hypertension | 149 (60.8) | 157 (61.8) | 0.8196 | 98 (57) | 51 (70) | 0.059 | 120 (58.8) | 37 (74) | 0.05 |
| Diabetes | 41 (16.7) | 57 (22.4) | 0.1087 | 29 (17) | 12 (16,4) | 0.935 | 42 (21) | 15 (30) | 0.15 |
| Stroke | 17 (6.9) | 9 (3.5) | 0.0880 | 10 (6) | 7 (10) | 0.285 | 4 (2) | 5 (10) | 0.01* |

* represents statistically significant p <0.05

## pQCT parameters and all-cause mortality

Among men, there was a significant decreasing risk of all-cause mortality with increasing quartiles of trabecular vBMD of radius and tibia, cortical vBMD of radius, cortical thickness of radius and tibia, and muscle density, Fig 1. All-cause mortality significantly increased with increasing quartiles of endosteal circumference of radius and tibia and across quartiles of SSIp at both the radius or tibia. Among women, all-cause mortality risk significantly decreased with increasing quartiles of cortical vBMD of radius and tibia, cortical thickness of radius and tibia, Fig 2. The mortality risk increased with the increasing quartiles of endosteal circumference of radius and tibia. All-cause mortality did not vary across quartiles of trabecular vBMD of radius or tibia, and muscle density. The Unadjusted Log Rank test p-value of the pQCT bone and muscle density quartiles and All-cause Mortality are shown in Table 2.

Among men, in multivariable adjusted continuous models, one standard deviation increase in trabecular and cortical vBMD at the radius and tibia, cortical thickness at both the radius and tibia, SSPi of the tibia (but not radius)and muscle density were all associated with significant lower mortality, Table 3.

One standard deviation increase in endosteal circumference at both the radius was associated with a 53–64% increase in mortality risk.

Among women, in multivariable adjusted continuous models, one standard deviation increase in cortical vBMD (but not trabecular vBMD) at both the radius and tibia, cortical thickness and SSPi were all associated with lower mortality, Table 4. Similar to findings in men, greater endosteal circumference was associated with an increased mortality. There was no association with muscle density and mortality in women.

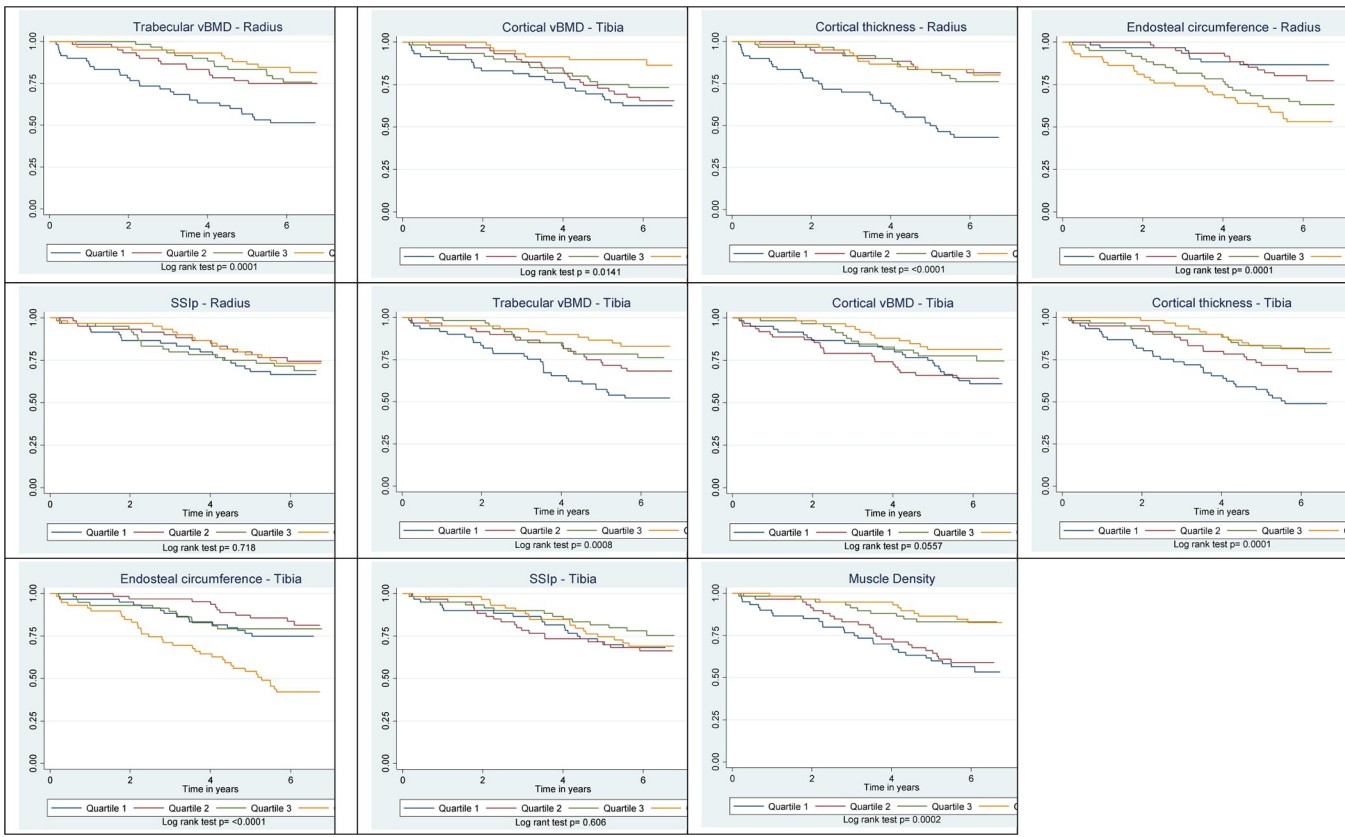

**Fig 1. Kaplan Meir curves of quartiles of pQCT parameters and log rank test in men.**

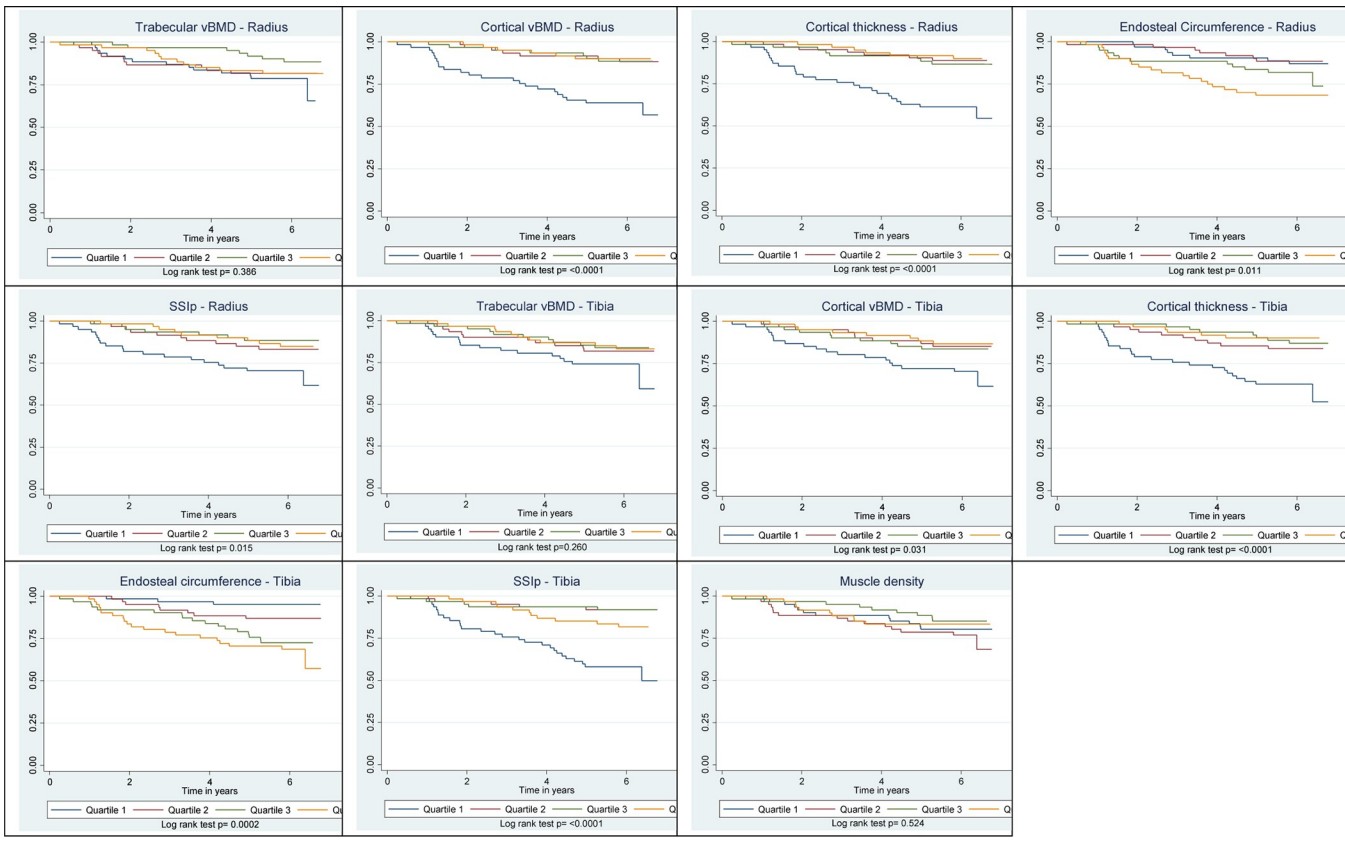

**Fig 2. Kaplan Meir curves of quartiles of pQCT parameters and log rank test in women.**

## Mediation analysis

The pQCT parameters which were significant in model 2 were compared with the mediator models (models 3 and 4) for identifying attenuation of hazards ratio of more than 10%. Among men, muscle density showed greater than 10% attenuation with the gait speed. Based

**Table 2. Unadjusted Log Rank test p-value of the pQCT bone and muscle density quartiles and all-cause mortality.**

| | Log rank test p value | |
| --- | --- | --- |
| **Variable** | **Men** | **Women** |
| Trabecular V BMD (Radius 4%) | 0.0001 | 0.39 |
| Cortical vBMD (Radius 33%) | 0.01 | <0.0001 |
| Cortical Thickness (Radius 33%) | <0.0001 | <0.0001 |
| Endosteal circumference (Radius 33%) | 0.0001 | 0.01 |
| SSI p (Radius 33%) | 0.72 | 0.02 |
| Trabecular V BMD (Tibia 4%) | 0.0008 | 0.26 |
| Cortical vBMD (Tibia 33%) | 0.06 | 0.03 |
| Cortical Thickness (Tibia 33%) | 0.0001 | <0.0001 |
| Endosteal circumference (Tibia 33%) | <0.0001 | 0.0002 |
| SSI p (Tibia 33%) | 0.61 | <0.0001 |
| Muscle Density | 0.0002 | 0.52 |

**Table 3. Hazard ratios (95% CI) for mortality per SD increase in the pQCT measures among 245 men of MILES.**

|  | Model 1 | | Model 2 | | Model 3 | | Model 4 | |
|---|---|---|---|---|---|---|---|---|
|  | HR | p value | HR | p value | HR | p value | HR | p value |
| Trabecular V BMD (Radius 4%) | 0.63 (0.47, 0.8) | 0.0018 | 0.59 (0.43, 0.81) | 0.0011 | 0.60 (0.44, 0.81) | 0.0009 | 0.59 (0.42, 0.82) | 0.0016 |
| Cortical vBMD (Radius 33%) | 0.63 (0.49, 0.81) | 0.0003 | 0.61 (0.47, 0.79) | 0.0003 | 0.62 (0.47, 0.80) | 0.0003 | 0.60 (0.46, 0.80) | 0.0004 |
| Cortical Thickness (Radius 33%) | 0.56 (0.43, 0.73) | <0.0001 | 0.55 (0.42, 0.72) | <0.0001 | 0.58 (0.44, 0.77) | 0.0001 | 0.52 (0.39, 0.69) | <0.0001 |
| Endosteal circumference (Radius 33%) | 1.56 (1.21, 2.00) | 0.0005 | 1.63 (1.25, 2.12) | 0.0003 | 1.56(1.20, 2.02) | 0.001 | 1.72 (1.32, 2.25) | <0.0001 |
| SSI p (Radius 33%) | 0.85 (0.65, 1.11) | 0.23 | 0.85 (0.64, 1.12) | 0.25 | 0.88 (0.66, 1.16) | 0.36 | 0.86 (0.65, 1.15) | 0.863 |
| Trabecular V BMD (Tibia 4%) | 0.64 (0.48, 0.853) | 0.002 | 0.60 (0.45, 0.81) | 0.001 | 0.64 (0.48, 0.86) | 0.003 | 0.62 (0.45, 0.86) | 0.0036 |
| Cortical vBMD (Tibia 33%) | 0.64 (0.51, 0.80) | 0.0001 | 0.62 (0.49, 0.79) | 0.0001 | 0.60 (0.47, 0.77) | <0.0001 | 0.63 (0.49, 0.80) | 0.0002 |
| Cortical Thickness (Tibia 33%) | 0.61 (0.48, 0.77) | <0.0001 | 0.60 (0.47, 0.77) | <0.0001 | 0.65 (0.51, 0.84) | 0.001 | 0.63 (0.49, 0.81) | 0.0004 |
| Endosteal circumference (Tibia 33%) | 1.54 (1.20, 1.96) | 0.0006 | 1.54 (1.19, 1.98) | 0.0009 | 1.5 (1.15, 1.95) | 0.003 | 1.45 (1.14, 1.94) | 0.0038 |
| SSI p (Tibia 33%) | 0.74 (0.55, 0.99) | 0.04 | 0.73 (0.54, 0.98) | 0.04 | 0.77 (0.57, 1.03) | 0.085 | 0.75 (0.56, 1.01) | 0.058 |
| Muscle Density[a] | 0.63 (0.49, 0.81) | 0.0003 | 0.67 (0.51, 0.87) | 0.003 | 0.73 (0.55, 0.96) | 0.025 | 0.78 (0.58, 1.05) | 0.095[$] |

Model 1 = Age, height and weight adjusted; Model 2 = Model 1 + current smoker, alcohol consumption, opinion of health status, ADL, log 25-hydroxy vitamin D, Hypertension, diabetes and stroke; Mediator models: Model 3 = Model 2 + grip strength; Model 4 = Model 2+Gait speed.

a = in models with muscle density as a predictor muscle cross sectional area was also included

$ represents attenuation of >10% comparing model 2 and model 3

& represents attenuation of >10% comparing model 2 and model 4

on the mediation analysis (not shown) gait speed mediates the association between muscle density and all-cause mortality significantly to the extent of 10% (95% CI: 0.2%, 19.7%).

Among women, attenuation of more than 10% in the grip strength model was observed for cortical vBMD at radius and tibia, cortical thickness at tibia, and SSIp at radius and tibia. In the gait speed model, attenuation of more than 10% was observed for cortical vBMD at radius and tibia, cortical thickness at radius and tibia and SSIP at tibia. However, the mediation analysis of these pQCT parameters observed no significant mediation by gait speed or grip strength on all-cause mortality among women.

**Table 4. Hazard ratios (95% CI) for mortality per SD increase in the pQCT bone measures and muscle density among 254 women of MILES.**

|  | Model 1 | | Model 2 | | Model 3 | | Model 4 | |
|---|---|---|---|---|---|---|---|---|
|  | HR | P value | HR | P value | HR | P value | HR | P value |
| Trabecular V BMD (Radius 4%) | 0.97(0.70,1.34) | 0.84 | 0.86(0.61,1.23) | 0.40 | 1.07(0.72,1.58) | 0.73 | 0.97(0.68,1.37) | 0.86 |
| Cortical vBMD (Radius 33%) | 0.58(0.42,0.7) | 0.0006 | 0.64(0.47,0.87) | 0.003 | 0.68(0.49,0.95) | 0.03 | 0.73(0.55,1.0) | 0.05 |
| Cortical Thickness (Radius 33%) | 0.49(0.32,0.73) | 0.0004 | 0.54(0.37,0.79) | 0.002 | 0.50(0.31,0.81) | 0.005 | 0.60(0.40,0.88) | 0.009 |
| Endosteal circumference (Radius 33%) | 1.40(1.01,1.93) | 0.04 | 1.33(0.97,1.82) | 0.07 | 1.44(0.99,2.09) | 0.056 | 1.29(0.91,1.82) | 0.15 |
| SSI p (Radius 33%) | 0.71(0.50,1.02) | 0.06 | 0.70(0.48,1.01) | 0.05 | 0.75(0.49,1.14) | 0.18 | 0.68(0.47,0.97) | 0.04 |
| Trabecular V BMD (Tibia 4%) | 0.95(0.69,1.30) | 0.74 | 0.91(0.66,1.28) | 0.60 | 1.12(0.77,1.63) | 0.55 | 1.00(0.73,1.39) | 0.96 |
| Cortical vBMD (Tibia 33%) | 0.64(0.48,0.87) | 0.004 | 0.60(0.45,0.79) | 0.0004 | 0.67(0.50,0.91) | 0.011 | 0.66(0.50,0.88) | 0.004 |
| Cortical Thickness (Tibia 33%) | 0.43(0.30,0.61) | <0.0001 | 0.43(0.30,0.61) | <0.0001 | 0.47(0.33,0.68) | <0.0001 | 0.50(0.35,0.71) | 0.0001 |
| Endosteal circumference (Tibia 33%) | 1.79(1.34,2.39) | <0.0001 | 1.83(1.37,2.45) | <0.0001 | 1.83(1.34,2.48) | 0.0001 | 1.77(1.32,2.38) | 0.0001 |
| SSI p (Tibia 33%) | 0.60(0.38,0.94) | 0.02 | 0.581(0.374,0.90) | 0.02 | 0.78(0.49,1.24) | 0.30 | 0.68 (0.44,1.03) | 0.07 |
| Muscle Density | 0.81(0.60,1.10) | 0.18 | 0.84(0.61,1.15) | 0.27 | 0.78(0.56,1.09) | 0.15 | 0.94(0.67,1.34) | 0.74 |

Model 1 –Age, height and weight adjusted; Model 2- Model 1 + current smoker, alcohol consumption, opinion of health status, ADL, Hypertension, diabetes and stroke; Model 3 –Model 2 + grip strength; Model 4 –Model 2+Gait speed.

Grey shaded are the statistically significant P <0.0011 comparison of characteristics

## Discussion

We showed that trabecular vBMD (radius and tibia), cortical vBMD (radius and tibia), cortical thickness (radius and tibia), endosteal circumference (radius and tibia), SSIp (tibia) and muscle density were independent predictors of all-cause mortality among rural south Indian older men. Cortical vBMD (radius and tibia), cortical thickness (radius and tibia), SSIp (radius and tibia) and endosteal circumference (radius and tibia) were independent predictors of all-cause mortality among women. Gait speed significantly mediated the association of mortality and muscle density among men. However, among women no significant mediation by gait speed or grip strength was observed with any of the pQCT parameters.

To our knowledge there have been no studies of pQCT derived bone measures and mortality. In the Age, Gene/Environment Susceptibility (AGES)—Reykjavik Study among men and women, QCT trabecular vBMD at proximal femur was inversely associated with all-cause mortality independent of covariates including gender but cortical vBMD was not associated with mortality [27]. In the African American-Diabetes Heart Study (AA-DHS), QCT was measured at chest and abdomen for thoracic and lumbar spine vBMD. Among men, thoracic and lumbar spine vBMD was inversely associated with all-cause mortality at lumbar vBMD [HR per SD increase = 0.70 (95% CI 0.52–0.95, p = 0.02)] and thoracic vBMD [HR per SD increase = 0.71 (95% CI 0.54–0.92, p = 0.01)], but no association was observed among women [28]. These findings were similar to our study wherein trabecular vBMD was associated with mortality among men but not among women.

It has been well established that fractures are associated with increased mortality [29]. Results from the (AGES)—Reykjavik Study showed that history of fracture before the bone assessment did not alter the vBMD and mortality association, whereas incident fracture after the bone assessment attenuated the mortality association with trabecular vBMD [27]. This suggests that fractures may explain the association between lower BMD and mortality. We did not have information on incident fractures but adjusting for history of fracture in the past five years had little effect on the association of trabecular vBMD and mortality among men and women; however, adjustment for fracture history attenuated the SSIp of radius association with mortality among women. Further studies are needed to understand the association and the role of incident fractures among older Indian population and mortality.

Previously, we showed that trabecular vBMD among Indian men was 1.3–1.5 SD units lower compared to Caucasian US men; cortical thickness was 0.8 to 1.2 SD units lower and endosteal circumferences 0.5–0.8 SD units higher among the Indian men. This may suggest that the Indian older population have lower bone density and strength measures that could influence earlier mortality either through fractures or through other mechanisms.

Trabecular bone loss is more pronounced at earlier ages before 50 years (women 37%, men 42%) compared to cortical bone loss [30]. Throughout adulthood, periosteal apposition counterbalances endosteal bone loss by reconfiguring the available bone mass to maintain biomechanical properties. However, with increasing age, bone loss shifts more to the cortical compartment leading to cortical thinning and increased cortical porosity that in turn leads to loss of biomechanical strength and increased risk for fracture [31]. Bone loss in aging is the net result of periosteal bone formation and endosteal bone resorption [32], however with increasing age, the bone resorption exceeds bone formation leading to bone loss in the endosteal region and an increase in the endosteal circumference. The lower levels of cortical vBMD, lower cortical thickness and higher endosteal circumference in our study among men and women, suggest increased cortical thinning and porosity that may impact mortality among the Indian older population. The lack of association of trabecular vBMD with mortality at least among women needs to be explored further.

The association between lower BMD and mortality may reflect common pathways. An association between cardiovascular disease (CVD) and BMD has been reported but the results of these studies have been inconclusive. A large analysis of NHANES III observed no association of low BMD and CVD mortality among men. However, among women soon after menopause, low BMD was associated with mortality from cardiovascular disease [33]. BMD was associated with mortality independent of coronary artery calcium score and chronic lung disease [27]. In a meta-analysis of 28 studies, low BMD was associated with an increased risk of developing coronary artery disease, cerebrovascular conditions, and CVD-associated death [34]. This may suggest that the CVD may share similar pathways in the link between BMD and mortality.

The association between low BMD and increased mortality may also reflect shared risk factors. Increased low grade inflammation has been linked to higher mortality [35, 36] and lower BMD [37, 38] and fractures [39, 40]. Endogenous sex hormones are associated with mortality [41, 42], lower BMD [43, 44] and fractures [45, 46]. Age at menopause is negatively associated with BMD [47] increased fractures [48, 49] and mortality [48] and Indian women have an earlier menopause [50, 51]. Nevertheless, we have no information on these factors and their influence on these bone-mortality relationships cannot be ruled out.

The mechanostat hypothesis and the concept of bone and muscle crosstalk suggest there may be association of bone and physical performance measures. pQCT bone measures have been associated with grip strength [52–60] and gait speed [57, 59, 60]. Grip strength [18] and gait speed [19] also predict mortality. Considering this, grip strength and gait speed may mediate the association between the pQCT bone measures and mortality. However, in our analysis we observed no mediation between the pQCT bone parameters and mortality.

Over a six year follow up of the inCHIANTI study in Italy, pQCT derived muscle density was associated with all-cause mortality (per SD increase, 0.78 [0.69–0.88]) in models adjusted for height and weight. This association was attenuated and was not significant in fully adjusted models [61]. These results are similar to our findings among women. Among older men in the MrOS with a mean follow up of 7.2 years, muscle density (pQCT derived) was significantly inversely associated with all-cause mortality independent of important covariates [62]. Among Afro-Caribbean men aged 40 years in the Tobago health study, muscle density was also significantly inversely associated with all-cause mortality when adjusted for age and other covariates [17]. In the AGES study, both higher thigh intramuscular and intermuscular fat was associated with an increased mortality in men; intramuscular fat but not intermuscular fat was associated with mortality in women. Overall, findings may differ in men than women perhaps reflecting higher testosterone levels. In general, these associations were much smaller in women compared to me [63]. Opportunistic analyses of abdominal pelvic CT scans revealed associations with a total muscle index and psoas muscle index and mortality in a small convenient sample of men and women [64]. The latter study did not stratify by sex. Our study also observed similar inverse relationship of muscle density and mortality among men but not among women. In our study, muscle density did not differ between men and women suggesting a potential gender influence on the adverse effects of fat infiltration into muscle.

Muscle density is a proxy measure of myosteatosis. Insulin resistance [65] and oxidative stress [66] are considered to be factors that influence myosteatosis and mortality. In our study, among men the association of muscle density and mortality persisted even after adjusting for diabetes status. This was similarly observed in other studies [17, 62] suggesting that there may be other pathways which influence the muscle density and mortality association. Gait speed is associated with muscle density [61, 67, 68] and is associated with mortality [19]. The mediation of gait speed between the muscle density and mortality was significant among men in our study, suggesting that gait speed may be in the pathway of myosteatosis and mortality.

The current analysis has several potential limitations. This was an observational study and thus causality cannot be determined. The study sample was small and was limited to a rural south Indian region but the magnitude of our associations were similar to previous reports from larger cohorts of European descent. As some of the key indicators of shared pathways were not measured, there could be residual confounding in these relationships. We were unable to look at cause specific deaths and future studies are needed to determine whether observed associations are driven by mortality from specific causes. As this was an exploratory and hypothesis generating study, we did not adjust for multiple comparisons. Finally, newer techniques to measure microarchitecture and cortical porosity, such as, high resolution pQCT are currently available but we had no access to this technology.

However, our study has some important strengths. To our knowledge this was the first study describing pQCT measures and mortality association among a unique population based random sample of older Indian men and women with excellent longitudinal follow. We adjusted for many important potential confounding variables and conducted mediation analysis.

In conclusion, this study presents unreported independent association of all-cause mortality with trabecular vBMD, cortical vBMD, cortical thickness, endosteal circumference, SSIp and muscle density among rural south Indian older men; and cortical vBMD, cortical thickness, endosteal circumference and SSIp among rural south Indian older women. Grip strength and gait speed did not mediate the association of bone and muscle among women; however, significant mediation was observed by gait speed on muscle density and mortality among men. Further research is needed to confirm our findings in larger Indian older populations and to study the role of mediation of some of the key factors that may underlie these associations.

## Supporting information

**S1 Data.**
(ZIP)

## Author Contributions

**Conceptualization:** Guru Rajesh Jammy, Jane A. Cauley.

**Data curation:** Guru Rajesh Jammy, Pawan Kumar Sharma.

**Formal analysis:** Guru Rajesh Jammy, Robert M. Boudreau.

**Funding acquisition:** Sudhakar Pesara Reddy, Anne B. Newman, Jane A. Cauley.

**Investigation:** Pawan Kumar Sharma, Sudhakar Pesara Reddy, Anne B. Newman, Jane A. Cauley.

**Methodology:** Guru Rajesh Jammy, Robert M. Boudreau.

**Project administration:** Sudhakar Pesara Reddy, Anne B. Newman, Jane A. Cauley.

**Resources:** Guru Rajesh Jammy, Pawan Kumar Sharma, Sudhakar Pesara Reddy.

**Software:** Guru Rajesh Jammy, Robert M. Boudreau.

**Supervision:** Sudhakar Pesara Reddy, Anne B. Newman, Jane A. Cauley.

**Validation:** Guru Rajesh Jammy, Robert M. Boudreau.

**Visualization:** Guru Rajesh Jammy, Robert M. Boudreau, Jane A. Cauley.

**Writing – original draft:** Guru Rajesh Jammy, Jane A. Cauley.

**Writing – review & editing:** Guru Rajesh Jammy, Robert M. Boudreau, Iva Miljkovic, Pawan Kumar Sharma, Sudhakar Pesara Reddy, Susan L. Greenspan, Anne B. Newman, Jane A. Cauley.

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
