## [Decision Letter · Decision Letter 0]

2 May 2022

PGPH-D-22-00199

Peripheral bone structure, geometry, and strength and muscle density as derived from peripheral quantitative computed tomography and mortality among rural south Indian older adults.

Dear Dr. Cauley,

Thank you for submitting your manuscript to PLOS Global Public Health. After careful consideration, we feel that it has merit but does not fully meet PLOS Global Public Health’s publication criteria as it currently stands. Therefore, we invite you to submit a revised version of the manuscript that addresses the points raised during the review process.

Please submit your revised manuscript by . If you will need more time than this to complete your revisions, please reply to this message or contact the journal office at globalpubhealth@plos.org. Please include the following items when submitting your revised manuscript:

We look forward to receiving your revised manuscript.

Kind regards,

Bert B. Little, MA, PhD, FAAAS, FRAI, FRSM, FRSPH

Academic Editor

Journal Requirements:

1. Please provide a/amend your detailed Financial Disclosure statement. This is published with the article. It must therefore be completed in full sentences and contain the exact wording you wish to be published.

- State the initials, alongside each funding source, of each author to receive each grant.

2. Please ensure that Funding Information and Financial Disclosure Statement are matched.

Additional Editor Comments (if provided):

Two expert reviewers have evaluated your manuscript.

Both agree that it should be published.

One reviewer suggests a reasonable modification to the analyses done.

The second reviewer has five minor concerns that should be easily addressed.

We look forward to you revised manuscript.

Reviewers' comments:

Reviewer's Responses to Questions

**Comments to the Author**

1. Does this manuscript meet PLOS Global Public Health’s publication criteria? Is the manuscript technically sound, and do the data support the conclusions? The manuscript must describe methodologically and ethically rigorous research with conclusions that are appropriately drawn based on the data presented.

Reviewer #1: Yes

Reviewer #2: Yes

2. Has the statistical analysis been performed appropriately and rigorously?

Reviewer #1: Yes

Reviewer #2: Yes

3. Have the authors made all data underlying the findings in their manuscript fully available (please refer to the Data Availability Statement at the start of the manuscript PDF file)?

Reviewer #1: Yes

Reviewer #2: No

4. Is the manuscript presented in an intelligible fashion and written in standard English?

Reviewer #1: No

Reviewer #2: Yes

5. Review Comments to the Author

Reviewer #1: Drs. Jammy and Cauley evaluated association between pQCT measures and mortality among rural Indian older adults. This study used the data from MILES study that was aimed to understand prevalence of chronic diseases and disability among this Indian population.

The manuscript is well written and I have no major concerns. Few minor comments are as follow:

1. Sample size and events were smaller to draw firm conclusion.

2. This is an exploratory analysis from the MOBILE study data and therefore, findings should be considered as hypothesis generating.

3. pQCT does not provide much details on trabecular microarchitecture or cortical porosity. It is limited by the resolution. The use of technique provides only limited mechanistic information. However, I understand that HR-pQCT may not be available in India. This should be discussed as a potential limitation.

4. It is not surprising that people with multiple comorbidities would have lower vBMD especially at distal radius and have higher risk for death probably from other comorbidities rather than fracture per se. The findings are expected but has limited clinical implication.

5. In this study, 25% of participant has diabetes. Assuming older age, most likely these participant would have T2D. Studies in Caucasian T2D have showed higher aBMD by DXA and higher trabecular thickness at the distal radius. It is interesting that’s not the case with these participants. I guess T2D phenotype and skeletal may be different in Caucasian vs Indians. Indian phenotype is considered as thin but higher visceral adiposity and this may have some adverse effect on skeletal health (Curr Opin Endocrinol Diabetes Obes. 2021 Aug 1;28(4):383-389).

Reviewer #2: Given that Model 1 (age, height, and weight) is little modified by the addition of covariates included in Models 2, 3, and 4, I suggest that they be removed. Greater emphasis should be placed on the fact that muscle density is a predictor only for men.

6. PLOS authors have the option to publish the peer review history of their article (what does this mean?). If published, this will include your full peer review and any attached files.

**Do you want your identity to be public for this peer review?** For information about this choice, including consent withdrawal, please see our Privacy Policy.

Reviewer #1: No

Reviewer #2: No

---

## [Editor Report · Decision Letter 1]

11 Jul 2022

Peripheral bone structure, geometry, and strength and muscle density as derived from peripheral quantitative computed tomography and mortality among rural south Indian older adults.

PGPH-D-22-00199R1

Dear Cauley,

We are pleased to inform you that your manuscript 'Peripheral bone structure, geometry, and strength and muscle density as derived from peripheral quantitative computed tomography and mortality among rural south Indian older adults.' has been provisionally accepted for publication in PLOS Global Public Health.

Best regards,

Bert B. Little, MA, PhD, FAAAS, FRAI, FRSM, FRSPH

Academic Editor
